# Mesoporous Activated Biochar from Crab Shell with Enhanced Adsorption Performance for Tetracycline

**DOI:** 10.3390/foods12051042

**Published:** 2023-03-01

**Authors:** Jiaxing Sun, Lili Ji, Xiao Han, Zhaodi Wu, Lu Cai, Jian Guo, Yaning Wang

**Affiliations:** 1National Marine Facilities Aquaculture Engineering Technology Research Center, Zhejiang Ocean University, Zhoushan 316022, China; 2College of Food and Pharmacy, Zhejiang Ocean University, Zhoushan 316022, China; 3Institute of Ocean Higher Education, Zhejiang Ocean University, Zhoushan 316022, China

**Keywords:** crab shell, activated biochar, tetracycline, adsorption mechanism

## Abstract

In this study, three mesoporous-activated crab shell biochars were prepared by carbonation and chemical activation with KOH (K−CSB), H_3_PO_4_ (P−CSB), and KMnO_4_ (M−CSB) to evaluate their tetracycline (TC) adsorption capacities. Characterization by SEM and a porosity analysis revealed that the K−CSB, P−CSB, and M−CSB possessed a puffy, mesoporous structure, with K−CSB exhibiting a larger specific surface area (1738 m^2^/g). FT-IR analysis revealed that abundant, surface ox-containing functional groups possessed by K−CSB, P−CSB, and M−CSB, such as −OH, C−O, and C=O, enhanced adsorption for TC, thereby enhancing their adsorption efficiency for TC. The maximum TC adsorption capacities of the K−CSB, P−CSB, and M−CSB were 380.92, 331.53, and 281.38 mg/g, respectively. The adsorption isotherms and kinetics data of the three TC adsorbents fit the Langmuir and pseudo-second-order model. The adsorption mechanism involved aperture filling, hydrogen bonding, electrostatic action, π-π EDA action, and complexation. As a low-cost and highly effective adsorbent for antibiotic wastewater treatment, activated crab shell biochar has enormous application potential.

## 1. Introduction

In spite of being utilized to treat biological bacterial infections [1,2], antibiotics can also be used as biological additives and growth promoters. Tetracycline (TC) is widely used as a broad-spectrum antibiotic due to its high quality, low cost, high antibacterial activity, and limited side effects [3,4,5]. Nevertheless, humans, livestock, and poultry are unable to absorb TC antibiotics completely, resulting in 70–90% of the antibiotics being excreted in feces and urine, causing environmental pollution [6,7,8,9]. In addition, the fact that TC is highly water soluble and chemically stable makes it easy for it to accumulate in the water environment [10,11,12], threatening human health and the environment’s ecology. Apart from inducing the transfer of bacterial resistance genes into the human body, TCs can enter the body and induce bone marrow suppression, hemolytic anemia, visual neuritis, or peripheral neuritis [13]. Tetracycline is one of the most common antibiotic contaminants in the water environment. It is used on a large scale in the aquaculture and medical industries. Concentrations in shallow groundwater have been measured at 184.2 ng/L [14]. The content of TC in wastewater discharged from pharmaceutical factories and aquaculture farms is higher than the content in domestic wastewater [15]. Consequently, eliminating residual TCs from the environment has become a research focus.

In recent years, numerous techniques for removing TCs from aqueous solutions have been developed. These include the microbial method [16], chemical oxidation [17], photocatalytic oxidation [18], physical separation [19], and adsorption [20]. Among these techniques, the application of microbial methods is limited due to the difficulty of biodegrading stable TC [21]. Chemical oxidation is an expensive and energy-intensive process. Physical separation is inefficient and prohibitively expensive. The adsorption technique has been extensively studied due to its high efficiency, low cost, and simple operation [22].

The type of adsorbent determines the adsorption efficiency. Various adsorbents have been developed for the adsorptive removal of TCs, such as biochar [23], mineral materials, nanomaterials [24,25], and metal skeletal organics [21].

Biochar, a carbon-rich byproduct produced by pyrolysis under anaerobic or limited oxygen conditions, possesses a large surface area, a rich pore structure, and various active groups [26].

Wang et. al [27] compared the effects of rice straw biochar and pig manure biochar on the removal of tetracycline from aqueous solutions. Their results indicated that the rice straw biochar, which was prepared at 600 °C, had a higher tetracycline adsorption capacity (14.185 mg/g) [27]. Biochar could also efficiently remove food and plant residues from aqueous solutions, demonstrating a maximum adsorption capacity of up to 15.2508 mg/g [28]. Conventional pyrolysis-produced biochars are limited in their specific surface area, pore volume, and functional groups, resulting in their relatively low actual TC removal in aqueous solutions.

Recently, an engineered biochar, modified to remove contaminants, has been proposed [29]. It is defined as a carbon-rich solid manufactured from biomass using pyrolysis technology in combination with modification techniques such as chemical, physical, or biological modification. This results in a higher specific surface area and adsorption capacity than conventional biochar. It also results in an increase in the removal of contaminants from wastewater [30]. Among the modification techniques, chemical modification typically employs acidic or basic chemicals to activate the biochar, increasing the availability of functional groups, cation exchange capacity, and surface area [31]. For instance, a lignin biochar activated by H_3_PO_4_ exhibited a superior adsorption performance with a maximum adsorption capacity of 475.48 mg/g for TC [32]. The biochar derived from aerobic granular sludge was modified by ZnCl_2_ (Zn-BC) to remove TC from an aqueous solution; its maximum TC adsorption performance was 93.44 mg/g [23]. Therefore, it is worthwhile to produce a high-performance engineered biochar as an adsorbent for further research.

The purpose of this study was to prepare a new crab shell biochar adsorbent through carbonization and chemical activation using KOH, H_3_PO_4_, and KMnO_4_. Its adsorption properties were characterized by FTIR, XRD, SEM, and BET. The mechanism of adsorption was analyzed accordingly. This study provides a new and promising research proposal for the treatment of antibiotic-contaminated wastewater using crab shell biosorbents.

## 2. Materials and Methods

### 2.1. Materials and Chemical Reagents

The crab shell used in the experiment was obtained from the Zhoushan seafood market, washed with tap water multiple times, dried overnight at 70 °C, and crushed through a 100 mesh sieve (ASTM standard). KOH, H_3_PO_4_, KMnO_4_ were purchased from Sinopharm Chemical Reagent Co., Ltd. (Shanghai, China), while TC hydrochloride (purity > 96%) was purchased from Sinopharm Chemical Reagent Co., Ltd. This experiment utilized only analytical-grade chemical reagents.

### 2.2. Preparation of Crab Shell Biochar

The pretreated crab shell was placed in a crucible and pyrolyzed in a tube furnace. Pre-pyrolysis, N_2_ was introduced into the tube furnace for 30 min to ensure an inert atmosphere. The pretreated crab shell was then calcined at 500, 600, 700, 800, and 900 °C for 2 h with a heating rate of 10 °C/min and an N_2_ flow rate of 0.25 L/min before being cooled to room temperature. The prepared crab shell biochar varieties are denoted as 500−CSB, 600−CSB, 700−CSB, 800−CSB, and 900−CSB.

### 2.3. Preparation of Modified Crab Shell Biochar

Specifically, 800−CSB was modified with KOH, H_3_PO_4_, and KMnO_4_ to produce an engineered crab shell biochar. The specific procedures are as follows.

KOH modification: The 800−CSB was thoroughly mixed with solid KOH (CSB/KOH (*w*/*w*) = 1:3). The the mixture was calcined at 800 °C for 2 h at a heating rate of 10 °C/min and an N_2_ flow rate of 0.25 L/min. After bringing the sample to room temperature, it was washed with deionized water until the pH was neutral, dried in an oven at 85 °C for 12 h, and sieved through a 100 mesh screen (ASTM standard). K−CSB stands for KOH-modified crab shell biochar.

H_3_PO_4_ modification: An amount of 5 g 800−CSB was immersed in 50 mL of a 20% H_3_PO_4_ solution for 24 h under magnetic agitation. The resulting solution was washed with deionized water until the pH was neutral, dried in an oven at 85 °C for 12 h, and sieved through a 100 mesh screen (ASTM standard). The biochar derived from crab shells modified with H_3_PO_4_ is designated as P−CSB.

KMnO_4_ modification: An amount of 5 g 800−CSB was immersed in 50 mL 100 g/L KMnO_4_ solution for 24 h under magnetic agitation. The resulting solution was washed with deionized water until the pH was neutral, dried in an oven at 85 °C for 12 h, and sieved through a 100 mesh screen (ASTM standard). KMnO_4_, also known as modified crab shell biochar, was called M−CSB.

### 2.4. Characterization of Crab Shell Biochar

A scanning electron microscope (SU8010, Hitachi, Tokyo, Japan) was used to examine the morphology and microstructure of the as-prepared samples. The N_2_ adsorption–desorption isotherm was determined by a static volumetric adsorption analyzer (Micromeritics ASAP 2460, Norcross, GA, Micromeritics USA) at 573K. The specific surface area and pore size distribution of the prepared samples were evaluated based on Brunauer–Emmett–Teller (BET) and Barret–Joyner–Halenda (BJH). The phase composition was analyzed by X-ray diffraction (XRD, D8 Advance, Bruker, Ettlingen, Germany) in the 10–80° angle range, and the functional groups were analyzed by Fourier transform infrared spectroscopy (FTIR, Nicolet 6700, Thermo Fisher Scientific, Waltham, MA, USA).

### 2.5. Batch Adsorption Experiments

Using modified crab shell biochars, the effects of as-prepared adsorbent dose, initial pH, TC concentration, and adsorption time on the TC adsorption performance were investigated. In the following experiments, the temperature was 25 °C, a 250 mL beaker was selected, and the stirring speed was 500 rpm. The adsorbent (K−CSB, P−CSB, and M−CSB) doses were 0.01, 0.03, 0.05, 0.07, and 0.09 g, and the initial pH values was adjusted to 3.0, 5.0, 7.0, 9.0, and 11.0, respectively, using 0.1 mol/L HCl or 0.1 mol/L NaOH. The TC concentration was within 100–500 mg/L, and the adsorption time was 0–3600 min for the batch adsorption experiments. After adsorption, the reaction system was centrifuged at 3800 r/min for 5 min, the supernatant was collected, and its absorbance value (OD, UV 2600, Shimadzu, Japan) was measured at 360 nm. The TC concentration (*C_0_* and *C_e_*) in the adsorption system was calculated. The adsorption rate (*R*) and equilibrium adsorption capacity of modified the CSBs for TC (*q_e_*) were determined from formulas in Equation (1) and Equation (2), respectively.
(1)$$q_{e}=\frac{(C_{0}-C_{e})V}{m}\;$$
(2)$$R\left(\%\right)=\frac{C_{0}-C_{e}}{C_{0}}\times 100\%$$
where *q_e_* (mg/g) is the adsorption capacity of modified CSBs for TC at equilibrium, C_0_ (mg/L) and *C_e_* (mg/L) are the initial and at-equilibrium TC concentrations, *V* (mL) is the volume of the TC solution, *m* (g) is the amount of modified crab shell biochar, and *R* (%) represents the TC adsorption rate.

### 2.6. Adsorption Kinetics and Isothermic Studies

In the adsorption kinetic experiments, 0.05 g of as-prepared adsorbent was added to 50 mL of TC solution with concentrations of 100, 200, and 400 mg/L, respectively. The adsorption capacity of the modified crab shell biochars for TC was calculated at different adsorption times, ranging from 0 to 2880 min, at 298 K. The adsorption kinetics of TC on the modified crab shell biochar were analyzed by the pseudo-first-order (PFO) and pseudo-second-order (PSO) kinetics models in Equations (3) and (4) [33,34].
(3)$$ln(q_{e}-q_{t})=lnq_{e}-K_{1}t\;$$
(4)$$\frac{t}{q_{t}}=\frac{1}{K_{2}q_{e}^{2}}+\frac{t}{q_{e}}\;\;$$
where *q_e_* (mg/g) and *q_t_* (mg/g) are the adsorption amounts of TC at equilibrium and at time *t* (min), respectively, and *K_1_* (min^−1^) and *K_2_* (g/mg min) are the rate constants of PFO and PSO, respectively.

In the adsorption isotherm experiments, 0.05 g of as-prepared adsorbent was added to 50 mL of a TC solution with initial concentrations of 50, 100, 150, 200, 300, 400, and 500 mg/L, respectively. After adsorption equilibrium, the adsorption capacity of the modified crab shell biochars for TC was calculated. The adsorption isotherm was analyzed by the Langmuir and Freundlich model in Equations (5) and (6) [35,36,37].
(5)$$\frac{c_{e}}{q_{e}}=\frac{c_{e}}{q_{m}}+\frac{1}{K_{L}q_{m}}\;$$
(6)$$lnq_{e}=lnK_{F}+\frac{1}{n}lnc_{e}\;$$
where *q_m_* (mg/g) is the maximum adsorption capacity of the modified crab shell biochars for TC, *q_e_* (mg/g) is the adsorption capacity at equilibrium, *C_e_* (mg/L) represents the TC concentration at equilibrium, *K_L_* (L/mg) is the Langmuir adsorption equilibrium constant, *K_F_* [(mg/g) (L/mg)^1/n^] is the Freundlich constant, and 1/n is the adsorption intensity factor or surface heterogeneity.

## 3. Results and Discussion

### 3.1. Characterization of Biochar

#### 3.1.1. Porous Structure

Table 1 shows the specific surface area, pore volume, and average pore size of the crab shell biochar prepared at different pyrolysis temperatures. Compared with the original crab shells, the specific surface areas of the crab shell biochar after high temperature pyrolysis are significantly higher, indicating that organic matter in the crab shells escapes and forms new pore structures in a high-temperature and oxygen-free environment. Moreover, with the increase in temperature, the specific surface area of the crab shell biochar firstly increases and then decreases, reaching a maximum at 800 °C. The variation trend of pore volume is consistent with that of specific surface area, while the variation trend of the average pore size is the opposite. This is because with the increase in the pyrolysis temperature, the organic matter in the crab shell gradually decomposes and forms new pores, increasing the specific surface area and pore volume. The pore could collapse or be blocked when the pyrolysis temperature is excessive; this affects the specific surface area and pore volume. In conclusion, 800−CSB can be selected as the original material for subsequent modifications according to the size of its specific surface area.

Figure 1 depicts the N_2_ adsorption and desorption isotherms of K−CSB, P−CSB, and M−CSB. According to the IUPAC classification, it could be demonstrated that their N_2_ adsorption isotherms are all Type IV, with irreversible adsorption and desorption isotherms. It can be seen from the N_2_ adsorption isotherms of the four samples that there is an obvious and sharp upward trend under low pressure. This is due to the monolayer adsorption of the micropores. It then rises slowly under medium pressure due to the capillary condensation of the adsorption liquid film on the pore wall, until it finally reaches the high-pressure region. In addition, the K−CSB isotherm (Figure 1a) shows a hysteresis loop at P/P_0_ > 0.4. This is consistent with the H4 hysteresis ring, indicating the presence of mesopores [38]. Furthermore, the pore size distribution (PSD) of the modified crab shell biochar was analyzed using a non-local density functional theory (NLDFT) model. The PSDs of K−CSB, P−CSB, and M−CSB are primarily in the range of 2–20 nm and are primarily composed of mesopores. The BET method analysis of the N_2_ adsorption isotherm data reveals that the specific surface area of 800−CSB is 181.79 m^2^/g. At the same time, the specific surface areas of K−CSB, P−CSB, and M−CSB after activation reached 1095.14, 381.16, and 271.25 m^2^/g, as shown in Table 2. It was demonstrated that KOH activation could produce materials with a greater specific surface area and pore volume. This is because at high temperatures, KOH is capable of reacting with the biochar C to produce K_2_CO_3_, K_2_O, and H_2_. K_2_CO_3_ and K_2_O can then react with C to form metallic K and CO, endowing its pore structure as metallic K enters the carbon interlayer and escapes the volatile gas [39].

#### 3.1.2. Morphological Analysis

Figure 2 depicts the microstructures of 800−CSB, K−CSB, P−CSB, and M−CSB at various magnifications. The structure of 800−CSB is relatively compact, with few openings and humps. Following the activation of the crab shell biochar, K−CSB, P−CSB, and M−CSB all display fluffy structures, with K−CSB displaying an irregular, layered structure with an abundance of micropores and mesopores, P−CSB displaying an etched, porous structure with abundant apertures on the surface, and M−CSB displaying a rough surface with numerous cracks. As activators, KOH, H_3_PO_4_, and KMnO_4_ can dehydrate or erode the carbohydrate shell of biochar in the chemical activation process to obtain a higher specific surface area and more surface functional groups [40]. This could provide more active sites for the adsorption of TCs.

#### 3.1.3. XRD Analysis

The X-ray diffraction analyses of the as-prepared samples were performed between 10° and 80° (2θ); Figure 3 depicts the XRD patterns. The 24.8° and 43.6° diffraction peaks correspond to amorphous C for the (002) and (100) crystal planes, respectively [41], and are all indexed by JCPDS card number 81-2030. The very sharp peaks of K−CSB indicate that C changes from amorphous to crystalline during the activation process, and the peaks at 29.4°, 35.9°, and 39.4° correspond to the (010), (110), and (11-3) crystal planes of graphite carbon (PDF#47-1743). Moreover, the diffraction peak at 32.6° is attributed to the (330) crystal plane of K_2_CO_3_ because KOH reacts with the C from the crab shell biochar to produce K_2_CO_3_ [42]. In addition, there are two prominent peaks at 12.8° (101) and 41.2° (220); these correspond to minerals containing primarily calcium and aluminum. Compared to the XRD pattern of 800−CSB, the patterns of P−CSB and M−CSB are unchanged. However, M−CSB displays a new diffraction peak at 36.1°, corresponding to the (211) crystal plane of Mn_3_O_4_ (JCPDS card No. 80-0382) [43] and indicating that MnO_x_ is loaded onto the surface of the crab shell biochar.

#### 3.1.4. FTIR Analysis

Figure 4 depicts the FTIR spectra of the samples as prepared. It can be demonstrated that the FTIR spectra of M−CSB and P−CSB are nearly identical to those of 800−CSB, while the spectra of KCSB vary significantly, which is consistent with the XRD results. The hydroxyl groups (−OH) in alcohol, phenol, and carboxylic acid are responsible for the adsorption peaks at 3440 cm^−1^ and 3381 cm^−1^ [44]. Those near 1600 cm^−1^ correspond to C=O stretching vibrations on the aromatic ring; those at 1096 cm^−1^ and 1061 cm^−1^ are due to C−O stretching vibrations on the ester, alcohol, ether, and acid [45]. Those near 600 cm^−1^ belong to the stretching vibration of the metal oxygen bond in the fingerprint region [46]. In addition, the adsorption peak located at 1388 cm^−1^ in the K−CSB spectra is the characteristic peak of K_2_CO_3_: at 2168 cm^−1^ it is ascribed to the stretching vibration C=C or C≡C, while at 881 cm^−1^ it is due to the bending vibration of C−H on the benzene ring. Furthermore, the adsorption peak at 1096 cm^−1^ in the P−CSB spectrum is due to the vibration of P=O [47], while the peak at 548 cm^−1^ in the M-CSB spectrum is the characteristic peak of MnO_X_. Thus, chemical activation endows crab shell biochar with an abundance of oxygen-containing functional groups, such as −OH, C−O, and C=O. These act as an π electron donor with an π electron acceptor (N− heteroaromatic ring or amino functional group) in TC, forming a π-π electron donor–acceptor (π-π EDA) [48] that is beneficial to the adsorption of TC.

### 3.2. Batch Adsorption Experiments

#### 3.2.1. Effect of Initial TC Concentration

The adsorption capacities of K−CSB, P−CSB, and M−CSB to TC with different initial concentrations vary with adsorption time, as depicted in Figure 5a, 5c, and 5d. It can be shown that the adsorption capacities of K−CSB, P−CSB, and M−CSB to TC initially spike and then gradually level off until they reach equilibrium. This is because there are numerous adsorption sites on the surface of the modified crab shell biochar at the beginning of adsorption; when the surface adsorption sites are filled, the TC begins to spread inward. This is a relatively slow process. In addition, it can be observed that as the initial concentration of TC rises, it takes longer to achieve equilibrium in the adsorption process. The effects of different initial TC concentrations on the equilibrium adsorption capacity (*q_e_*) and adsorption rate (*R*) of K−CSB, P−CSB, and M−CSB to TC are depicted in Figure 5b, 5d, and 5f, respectively. With an increase in the initial TC concentration, the equilibrium adsorption capacity (*q_e_*) of the K−CSB, P−CSB, and M−CSB to TC increases gradually and then levels off. In contrast, the adsorption rate (*R*) decreases. When the initial concentration of TC was 100 mg/L in the adsorption experiment, the adsorption rate of the samples was the highest (96.98%, 97.29%, and 91.79%). With the increase in the initial concentration of TC solution, the adsorption rates of the materials began to decline. When the initial concentration of TC was 500 mg/L, the adsorption rates of materials were 74.73%, 66.20%, and 52.93%, respectively.

For example, in a 400 mg/L TC solution, the maximum adsorption capacities of K−CSB, P−CSB, and M−CSB to TC are 380.92, 331.53, and 281.38 mg/g, respectively, and the corresponding adsorption rates are 95.23%, 82.89%, and 70.41%, respectively.

#### 3.2.2. Effect of Adsorbent Dose

Figure 6 depicts the effect of various adsorbent doses on the equilibrium adsorption capacity (*q_e_*) and adsorption rate (*R*) of K−CSB, P−CSB, and M−CSB to TC. As the adsorbent dose (K−CSB, P−CSB, and M−CSB) increases, the adsorption capacity (*q_e_*) demonstrates a general downward trend, initially declining rapidly and then tending to decline gradually. Although the increase in biochar dosage provides more adsorption sites for adsorption and increases the overall adsorption capacity, the total amount of TC adsorbed in the solution is fixed. The adsorption capacity of biochar dispersed to each unit mass will decrease when the dosage increases. As can be seen from Figure 6, when the dosage is 0.01–0.05 g, the adsorption rates of the three samples with respect to TC increase at the adsorption equilibrium with an increase in the dosage. When the dosage was 0.05 g, the adsorption rate reached its highest value and showed a slightly decreasing trend between 0.05–0.09 g. This is because the increase in biochar in a certain range can promote an increase in the adsorption rate. On one hand, the decrease occurs because the excessive dosage leads to a collision between the materials or the agglomeration of blocks, reducing the adsorption sites. On the other hand, the concentration of the TC solution decreases with the process of adsorption, and the interaction force between the TC and the biochar surface gradually decreases [49], reducing the adsorption rate of the materials.

When the material dosage is 0.05 g, the adsorption capacity of the modified crab shell biochars for the TC solution reaches its maximum. Therefore, the adsorption of 1 g/L TC of 50 mL is optimal when the dosage of K−CSB, P−CSB, and M−CSB is 0.05 g.

#### 3.2.3. Effect of Initial pH

Figure 7 illustrates the effect of the initial pH on the TC adsorption rate of K−CSB, P−CSB, and M−CSB. It can be demonstrated that the adsorption capacity for TC of modified CSBs initially gradually increases and then decreases as the initial pH increases, primarily due to the charge difference between the TC and the modified carbohydrate shell biochar under different pH conditions. Tetracycline antibiotics all contain a phenolic hydroxyl group, an enol hydroxyl group, and a dimethylamine group. These drugs are amphoteric compounds with three pKa values of 2.8~3.4, 7.2~7.8, and 9.1~9.7, respectively. [50]. At pH < 3.3, TC primarily exists as cations; at pH < 7.7, it exists as amphoteric ions and gradually changes from cations to neutral ions and anions; and at pH > 7.7, it primarily exists as an anion [51]. Additionally, the surface charge of the modified carbohydrate shell biochar varies with the pH. The H^+^ in the acidic solution protonates the biochar surface, giving it a positive charge; in the alkaline solution, the charge is negative [52]. When pH = 3–7, electrostatic repulsion occurs between the positively charged biochar and the cationic TC. As the pH increases, the TC changes from a cationic state to a neutral ion or anionic state, the repulsion gradually decreases, and the adsorption rate reaches its maximum at pH = 7. When the pH > 7, the repulsive interaction between the negatively charged biochar and the anionic TC gradually increases. Therefore, the optimal pH for TC adsorption by K−CSB, P−CSB, and M−CSB is 7.

#### 3.2.4. Effect of Adsorption Time

Figure 8 depicts the effect of adsorption time on the adsorption of TC by K−CSB, P−CSB, and M−CSB. In order to explore the influence of adsorption time on the adsorption effect of TC, an initial concentration of TC of 400 mg/L was used, and the sampling time lasted 3600 min. As the rich pore structure and functional groups on the surface of the as-prepared samples can provide many TC adsorption sites, the adsorption rates of the three TC adsorbents increases dramatically between 0 and 360 min. The optimal adsorption time for the three sorbents is 2880 min. In the 360–2880 min adsorption stage, the adsorption sites become progressively occupied, the TC begins to diffuse to the internal sites and functional groups of the biochar, and the adsorption rate continues to exhibit a slow growth trend. Moreover, between 2880 and 3600 min, the adsorption reaches an equilibrium or decreases slightly, indicating that equilibrium has been reached.

### 3.3. Adsorption Kinetics Studies

The experimental kinetic data for the adsorption kinetics of modified carbohydrate shell biochar for TC at different concentrations are fitted by PFO and PSO models, as shown in Figure 9. The adsorption kinetic parameters are shown in Table 3. It can be observed that the *R^2^* values of the PSO model are greater than those of the PFO model. Therefore, the PSO model best describes the adsorption of TC by K−CSB, P−CSB, and M−CSB. It used to calculate the adsorption amounts of TC at equilibrium in a 400 mg/L TC solution (384.62, 333.33, and 277.77 mg/g), which are in good agreement with experimental data (380.92, 331.53 and 281.38 mg/g). It indicates that the adsorption of TC on K−CSB, P−CSB, and M−CSB is a chemical process involving the exchange of electrons between functional groups. In addition, the fitting adsorption capacity (*q_e_*) increases as the initial TC concentration rises due to the intense competition among TC for the surface-active sites of biochar in a solution with a high TC concentration.

### 3.4. Adsorption Isotherm Studies

The Langmuir model and the Freundlich model of TC adsorption on K−CSB, M−CSB, and P−CSB were fitted by adsorption isotherm experiments, as depicted in Figure 10, and the adsorption isotherm model parameters are listed in Table 4.

The correlation coefficients *R^2^* of the Langmuir isotherm equation were 0.9852, 0.9933, and 0.9939, respectively, higher than those of Freundlich isotherm equation (*R^2^* = 0.6901, 0.8259, and 0.7208). This indicates that the adsorption of the modified crab shell biochars is more consistent with the Langmuir isothermal model distribution. Therefore, the adsorption of TC on K−CSB, P−CSB, and M−CSB is mainly monolayer adsorption.

The Langmuir isotherm equation is commonly used to evaluate the maximum adsorption capacity (*q_e_*) of adsorbed materials. Using the Langmuir equation, the equilibrium adsorption amounts of TC in a 400 mg/L TC solution are calculated to be 400.00, 357.14, and 277.78 mg/g, which agree with the experimental data (380.92, 331.53, and 281.38 mg/g).

## 4. Adsorption Mechanism

Experiments on batch adsorption demonstrated that adsorbents prepared from activated crab shell biochar have a superior TC adsorption performance. The adsorption performance of K−CSB is the best, followed by P−CSB and M−CSB. The results of adsorption kinetics and isotherms indicate that the adsorption process of TC by K−CSB, P−CSB, and M−CSB is chemical monolayer adsorption. At the same time, the abundant porous structure of the three adsorbents contributes to the adsorption of TC molecules by aperture filling, i.e., physical adsorption. The above is the mechanism of TC adsorption by three adsorbents. Chemical adsorption involves hydrogen bonding, electrostatic action, π-π EDA action, and complexation (Figure 11). These oxygen-containing functional groups on the biochar surface can protonate with H^+^ to form positively charged functional groups, which then combine electrostatically with negatively charged TC [53]. Moreover, these oxygen-containing functional groups, such as −OH, C−O, and C=O, can act as π electron donors with π electron acceptors (N-heteroaromatic ring or amino functional group) in TC, forming an π-π electron donor-acceptor. In addition, a high concentration of −OH on the surface may form hydrogen bonds with the amino group [54]. In addition, the XRD analysis results suggest that the mineral ions on the surface of activated crab shell biochar may facilitate the complexation-based adsorption of TC.

## 5. Conclusions

In this study, waste crab shell was used as the original biomass material for preparing activated biochar with KOH, H_3_PO_4_, and KMnO_4_, endowing them with a greater specific surface area, a well-developed porosity, an abundance of oxygen-containing functional groups, and a vast number of bonding sites for TC molecules. In addition, the unique, mesoporous structures of the activated crab shell biochar contribute to its excellent adsorption performance. The maximum TC adsorption capacities of K−CSB, P−CSB, and M−CSB are 380.92, 331.53, and 281.38 mg/g, respectively. The Langmuir and pseudo−second−order model can better describe the adsorption of three kinds of adsorbents on TC, indicating that the adsorption process of K−CSB, P−CSB, and M−CSB on TC is dominated by chemical monolayer adsorption. Furthermore, the adsorption mechanism of TC by the three adsorbents comprises physical and chemical adsorption, i.e., aperture filling, hydrogen bonding, electrostatic action, π-π EDA action, and complexation. The results indicate that crab shells could serve as an innovative, cost-effective, and promising biosorbent for the treatment of the wastewater polluted by antibiotics.

## Figures and Tables

**Figure 1 foods-12-01042-f001:**
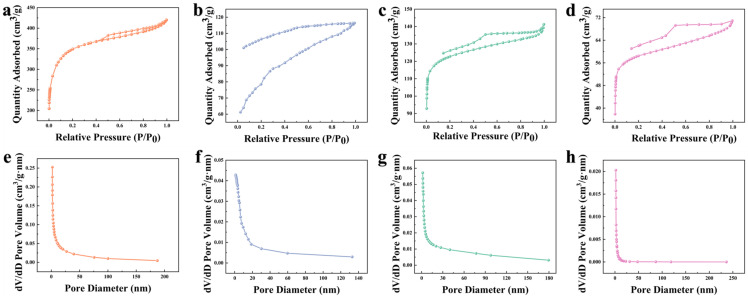
N_2_ adsorption–desorption isotherms and pore size distribution of the samples ((**a**,**e**) are K−CSB; (**b**,**f**) are P−CSB; (**c**,**g**) are M−CSB; and (**d**,**h**) are 800−CSB).

**Figure 2 foods-12-01042-f002:**
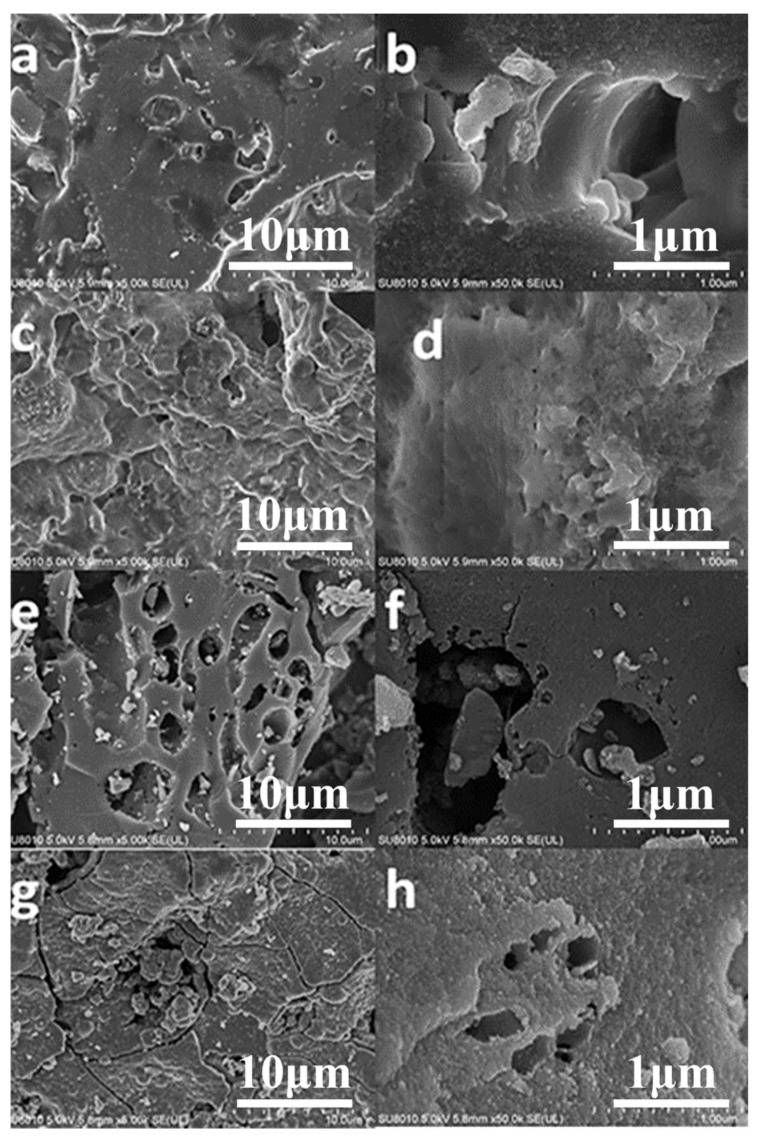
SEM images of crab shell biochar ((**a**,**b**) are 800−CSB; (**c**,**d**) are K−CSB; (**e**,**f**) are P−CSB; and (**g**,**h**) are M−CSB).

**Figure 3 foods-12-01042-f003:**
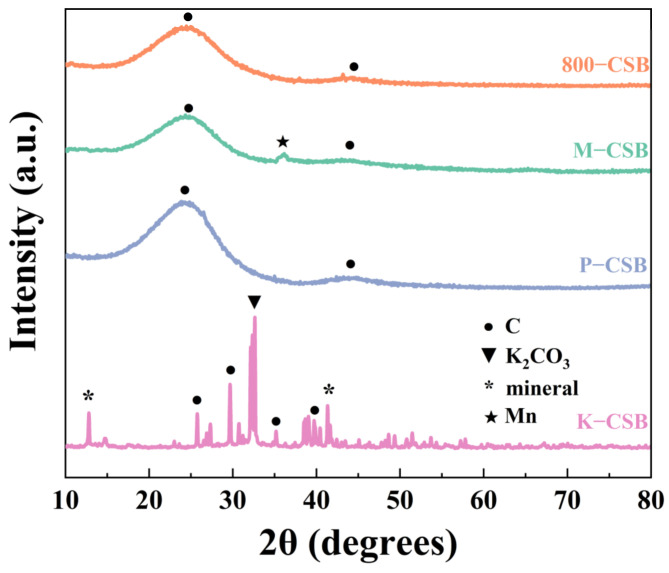
XRD pattern of modified biochar.

**Figure 4 foods-12-01042-f004:**
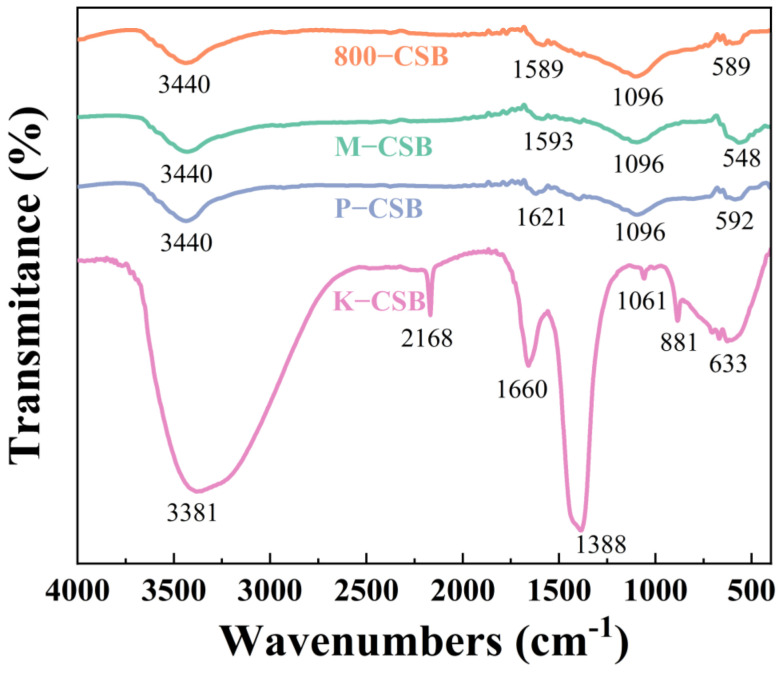
Infrared spectrum of crab shell biochar.

**Figure 5 foods-12-01042-f005:**
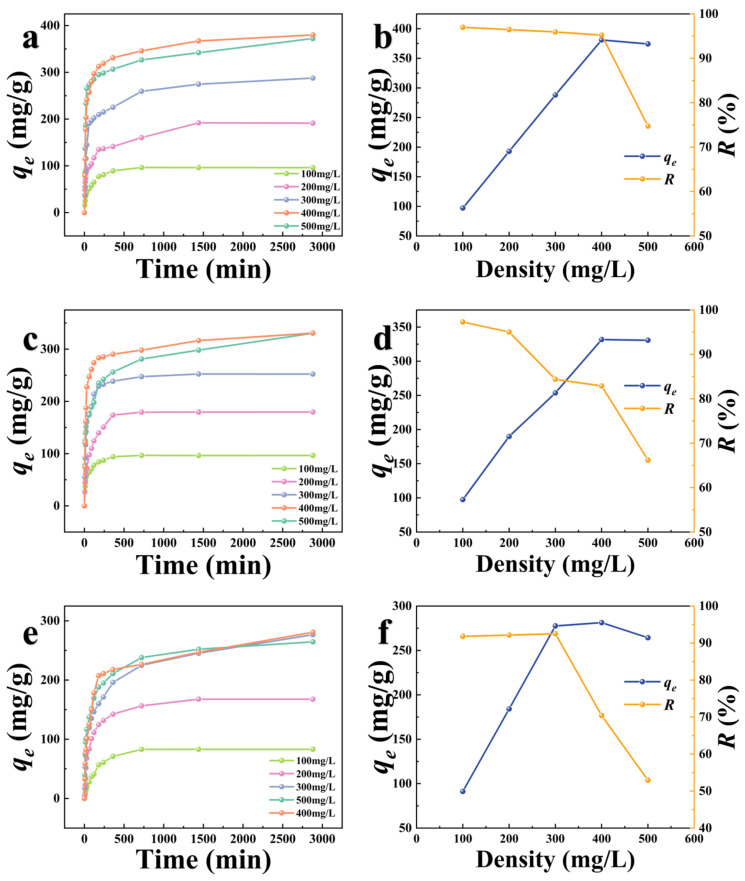
Adsorption of TC with different initial concentrations by modified carbon shell biochars at different times ((**a**,**b**) are K−CSB; (**c**,**d**) are P−CSB; and (**e**,**f**) are M−CSB).

**Figure 6 foods-12-01042-f006:**
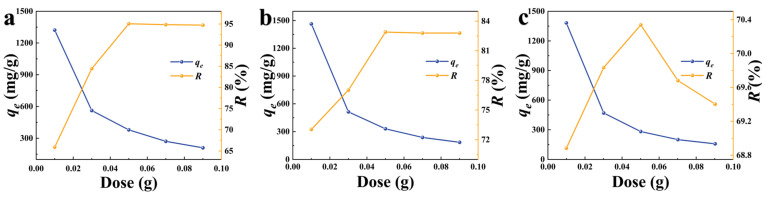
The effect of absorbent dosage on TC adsorption by modified crab shell biochar ((**a**) is K−CSB; (**b**) is P−CSB; and (**c**) is M−CSB).

**Figure 7 foods-12-01042-f007:**
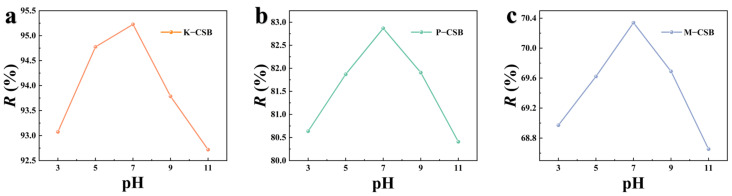
The effect of initial pH on TC adsorption by modified carb shell biochar ((**a**) is K−CSB, (**b**) is P−CSB, and (**c**) is M−CSB).

**Figure 8 foods-12-01042-f008:**
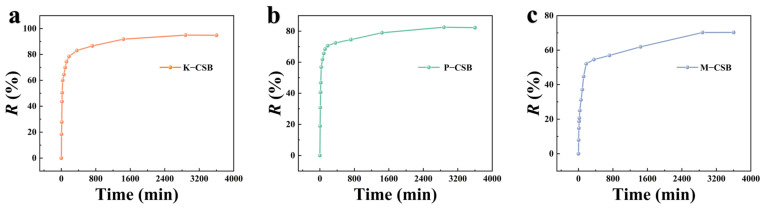
The effect of time on 400 mg/L TC adsorption by modified biochar ((**a**) is K−CSB; (**b**) is P−CSB; and (**c**) is M−CSB).

**Figure 9 foods-12-01042-f009:**
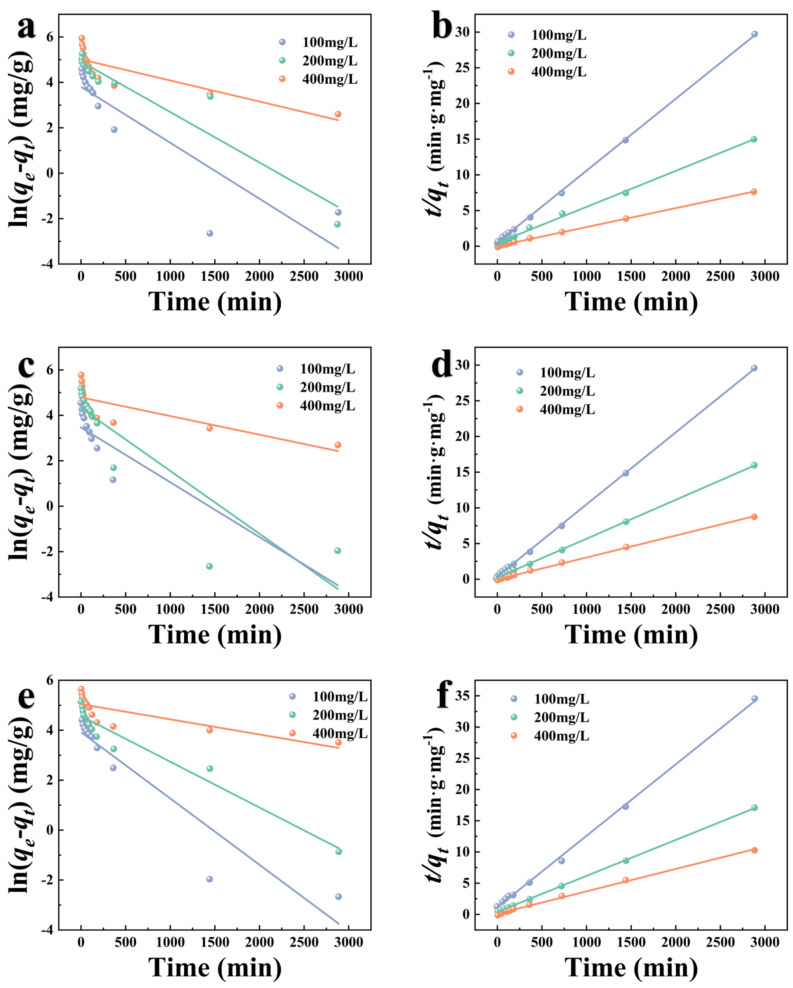
The adsorption kinetics models of TC adsorption on modified crab shell biochar ((**a**,**c**,**e**) are pseudo−first−order model of K−CSB, P−CSB and M−CSB, respectively; and (**b**,**d**,**f**) are pseudo−second−order model of K−CSB, P−CSB and M−CSB, respectively).

**Figure 10 foods-12-01042-f010:**
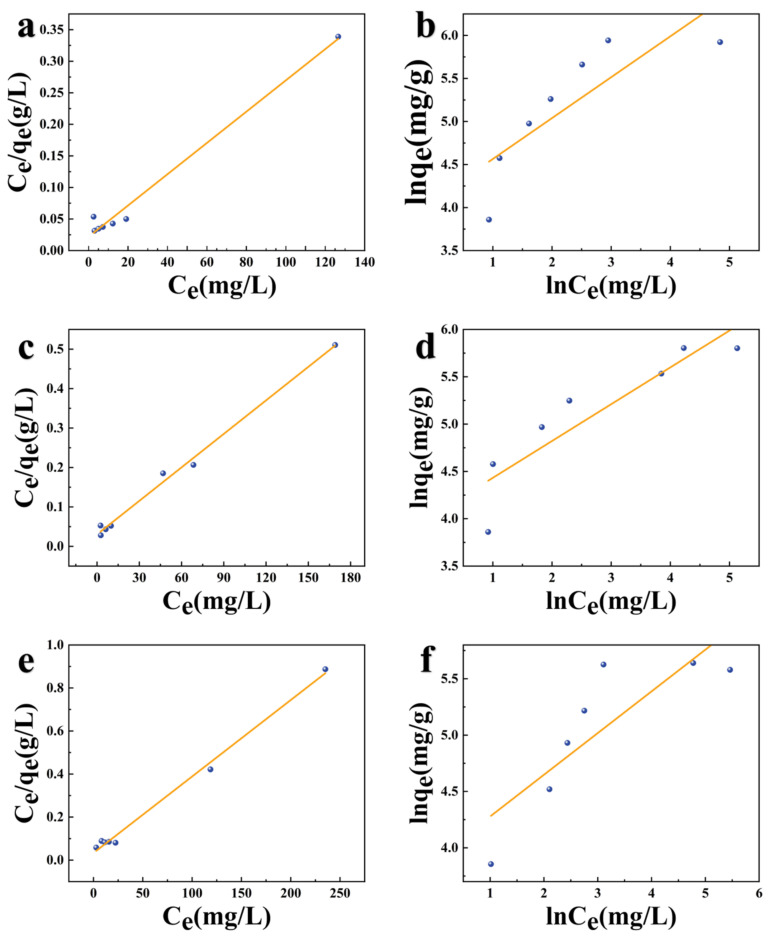
The adsorption isotherm models of TC adsorption on modified crab shell biochar ((**a**,**c**,**e**) are Langmuir isotherm models of K−CSB, P−CSB, and M−CSB, respectively; and (**b**,**d**,**f)** are Freundlich isotherm models of K−CSB, P−CSB, and M−CSB, respectively).

**Figure 11 foods-12-01042-f011:**
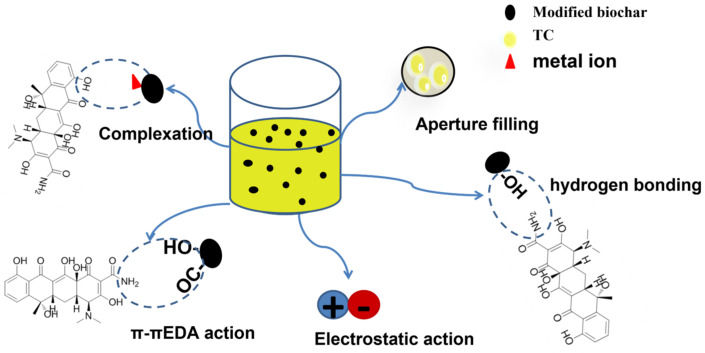
Mechanism diagram of TC adsorption on activated crab shell biochar.

**Table 1 foods-12-01042-t001:** BET parameters of crab shell biochar at different pyrolysis temperatures.

Sample	SBET (m^2^/g)	V_tot_ (cm^3^/g)	Average Pore Diameter (nm)
CS	9.57	0.03	10.84
500−CSB	41.16	0.02	2.26
600−CSB	78.11	0.04	2.05
700−CSB	127.38	0.05	1.68
800−CSB	181.79	0.11	2.37
900−CSB	177.91	0.11	2.41

**Table 2 foods-12-01042-t002:** Texture characteristics of the samples.

Sample	SBET (m^2^/g)	V_tot_ (cm^3^/g)	Average Pore Diameter (nm)
800−CSB	181.79	0.11	2.37
M−CSB	271.25	0.18	2.66
K−CSB	1095.14	0.63	2.18
P−CSB	381.16	0.21	2.24

**Table 3 foods-12-01042-t003:** The adsorption kinetic parameters of TC adsorption on modified carb shell biochars.

Sample		PFO			PSO		
		*q_e_ _1_* (mg/g)	*K_1_* g/(mg·min)	*R^2^*	*q_e_ _2_* (mg/g)	*K_2_* g/(mg·min)	*R^2^*
K−CSB	100 mg/L	47.8322	0.0025	0.7955	98.0392	0.0003	0.9994
	200 mg/L	125.8134	0.0024	0.9874	196.0784	0.0001	0.9968
	400 mg/L	172.0526	0.0020	0.7737	384.6154	0.0001	0.9994
P−CSB	100 mg/L	36.4339	0.0025	0.7783	98.0392	0.0005	0.9998
	200 mg/L	81.5731	0.0028	0.8008	181.8182	0.0002	0.9995
	400 mg/L	130.5949	0.0018	0.6621	333.3333	0.0001	0.9992
M−CSB	100 mg/L	57.9511	0.0027	0.9046	86.9565	0.0001	0.9770
	200 mg/L	98.4944	0.0002	0.9287	172.4138	0.0001	0.9958
	400 mg/L	174.2690	0.0014	0.7280	277.7778	0.0001	0.9954

**Table 4 foods-12-01042-t004:** The adsorption isotherm model parameters of TC adsorption on modified carb shell biochar.

Sample	Langmuir			Freundlich		
	*q_m_* (mg/g)	*K_L_* (L/mg)	*R^2^*	*n*	*K_F_* [(mg/g)(L/mg)^1/n^]	*R^2^*
K−CSB	400.00	0.1142	0.9852	2.1044	59.7041	0.6901
P−CSB	357.14	5.5044	0.9933	2.5714	57.0484	0.8259
M−CSB	277.78	1.8072	0.9939	2.7034	49.8291	0.7208

## Data Availability

The data presented in this study are available on request from the corresponding author.

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
