# Peer review of "Mesoporous Activated Biochar from Crab Shell with Enhanced Adsorption Performance for Tetracycline"

_foods, 2023, doi:10.3390/foods12051042_

Round 1

Reviewer 1 Report

I respectfully present to the authors my comments on the work entitled,

Mesoporous Activated Biochar from Crab Shell with Enhanced Adsorption Performance for Tetracycline

This work presents an interesting study on the sorption of an antibiotic. Tetracycline (TC) should be removed from the water because of its toxic effects on living beings. Additionally, the authors compare the results obtained with the 800-CSB adsorbent, which is very suitable. It allows us to see if the treatments with KOH, H3PO4, and KMNO4 modify or not the structure of the original material.

My comments are the following:

2.5. Batch adsorption experiments 

The authors do not indicate the conditions under which the studies were performed (in batch flasks, solution volume, stirring speed, and temperature).

Figure 2

Indicate the number of magnifications

Figures 5-7

Please improve the quality of the images because they are small and pixelated when increasing size.

3.2.2. Effect of adsorbent dose 

The authors indicate: the adsorption rate increases rapidly and reaches its maximum at 0.05 g, so 1g/L of K-CSB, P-CSB, and M-CSB is optimal for TC adsorption.

Please indicate the volume of the solution to assert that the best concentration of the adsorbent is 1g/L.

3.2.3. Effect of initial pH 

Are initial TC concentrations 380.92 (K-CSB), 331.53 (P-CSB), and 281.38 mg/g (M-CSB)? 

3.2.4. Effect of adsorption time 

Which is the initial TC concentration and the initial solution pH?

3.3. Adsorption kinetics studies 

Was the solution pH tested at 7?

Lines 306-307

The authors mention: 

In addition, the fitting adsorption capacity (qe) increases as the initial dye concentration rises due to the intense competition among TC. 

Should it be the initial antibiotic concentration?

Line 332

The manuscript indicates adsorbents prepared from activated crab shell biochar have superior TC adsorption performance.

The authors should explain superior to which adsorbent or to what.

Author Response

Manuscript ID: foods-2173468
Title: Mesoporous Activated Biochar from Crab Shell with Enhanced Adsorption Performance for Tetracycline

Dear Reviewers,
We would like to thank for Foods for giving us the opportunity to revise our manuscript. And we also thank the reviewers for their careful read and thoughtful comments on previous draft. Their comments and guidance added a lot to the research and increased its scientific content. Therefore, words cannot express their gratitude for the time and effort they put into evaluating this work. We have carefully taken their comments into consideration in preparing our revision, which has resulted in a paper that is clearer, more compelling and broader. The following summarizes how we responded to reviewers’ comments.
Below is our response to their comments.
Thanks for all the help.
Best wishes,
Jiaxing Sun

Revision-author’s response
To Reviewer A:
1. 2.5. Batch adsorption experiments
The authors do not indicate the conditions under which the studies were performed (in batch flasks, solution volume, stirring speed, and temperature).
1A: Thanks for your suggestion, we have added experimental conditions in lines 126-127.
2. Figure 2
Indicate the number of magnifications
2A: Thank you for your suggestion. We have marked the magnifications of the Figure 2.
3. Figures 5-7.
Please improve the quality of the images because they are small and pixelated when increasing size.
3A: Thanks to your guidance, we have improved the pixel and size of the images.
4. 3.2.2. Effect of adsorbent dose
The authors indicate: the adsorption rate increases rapidly and reaches its maximum at 0.05 g, so 1g/L of K-CSB, P-CSB, and M-CSB is optimal for TC adsorption.
Please indicate the volume of the solution to assert that the best concentration of the adsorbent is 1g/L.
4A: We revised the manuscript in lines 286-288, adding the volume of the solution.
5. 3.2.3. Effect of initial pH
Are initial TC concentrations 380.92 (K-CSB), 331.53 (P-CSB), and 281.38 mg/g (M-CSB)?
5A: In the experiment on the effect of initial pH, the initial concentration of TC was 400 mg/L.
6. 3.2.4. Effect of adsorption time
Which is the initial TC concentration and the initial solution pH?
6A: In the experiment on the effect of adsorption time, the initial concentration of TC was 400 mg/L, the initial pH of the solution was 7.
7. 3.3. Adsorption kinetics studies
Was the solution pH tested at 7?
7A: In the experiment of adsorption kinetics, the pH of the solution was 7.
8. Lines 306-307
The authors mention:
In addition, the fitting adsorption capacity (qe) increases as the initial dye concentration rises due to the intense competition among TC.
Should it be the initial antibiotic concentration?
8A: Thanks for your guidance, we have changed the dye into TC.
9. Line 332
The manuscript indicates adsorbents prepared from activated crab shell biochar have superior TC adsorption performance.
The authors should explain superior to which adsorbent or to what.
9A: We added the comparison of three adsorbents, specifically in lines 368-369.

Reviewer 2 Report

foods-2173468

 Dear Authors,

 After reading the manuscript I can conclude that it is mostly understandable written and good conceived, however, some parts need serious revision.

The manuscript deals with removal of antibiotic tetracycline (TC) from aqueous solution using modified (activated) crab shell. Original crab shell was taken from the seafood market. Prior chemical modification (using KOH, H3PO4 and KMnO4) the original material was prepared by pyrolysis and calcination.

In general, the questions that authors should consider and answer in revised version of the manuscript are:

-          Why particularly those three chemicals are chosen as modifying agents? This should be explained in the manuscript. Also, the reason for choosing crab shell as original material. I suppose why, but this should be explained in the manuscript. Also, why the tetracycline is chosen among numerous of antibiotics used? It would be good to explain that choice - to give more information about consumption of tetracycline, maybe amounts on an annual level…

-          Moreover, most of the figures (graphs) in the manuscript need revision in a way to be enlarged so that all numbers and text in Figures be visible. Otherwise, they don’t have the purpose.

-          I don’t see the results with original crab shell so it is not clear is modification even justified? This must be included in the manuscript. Modification usually increase the removal efficiency, but it also causes the increased costs and possible secondary pollution, since new chemicals are introduced in the process. Due this, the modification must be justified. So, the discussion about this must be also included in the paper.

 Detail comments line by line:

 Abstract

Line 12 – The abbreviations K-CSB, P-CSB and M-CSB should be mentioned earlier in line 11, in the brackets after the full names, and then they can be used in this form through the manuscript.

Line 15 – is this absorption or adsorption?

 1.      Introduction

Line 28 – Is this antibiotic only used for livestock and poultry? It seems like that by reading this sentence. Is it also prescribed for people? If yes, is it mean that humans are capable to completely absorb TC?

Moreover, as mentioned in my general comments, the introduction section should be expanded with information about TC, for example why the tetracycline is chosen among numerous of antibiotics used; add information about consumption of tetracycline, maybe amounts on an annual level…

 Lines 50-51 – Please check and revise the sentence. Also, please add/give the reference on which you refer to with this sentence.

 Line 71 - …using instead of “of KOH, H3PO4 and KMnO4.”

 2.      Materials and Methods

 Line 79 - 100-mesh sieve – is it the same as mentioned in line 96? If yes, this should be equalized.

 Line 86 – Is there a specific and justifiable reason why calcination was done at 800 °C? It could be shortly explained in the text.

 Line 92 – Please give the information about KOH concentration.

Line 93 – The CSB was calcined twice at 800 °C, before and after modification? Can you shortly explain why?

 Line 112 – Authors should refer here at certain reference/references in regard to the BET method. In general, the main data about the methods applied for characterisation of crab shell biochar should be given here.

 Line 117 – Did you used also unmodified original CSB in adsorption studies? I don’t see these results. It is very important for comparison with chemically modified samples in order to determine is the modification justified (look general comments).

 Lines 126, 137 and 153 – capacity of TC – this must be corrected. It is not capacity of TC; it is capacity of modified CSBs for TC.

 Line 143 – in qt the t should be in subscript.

 Line 167 – please give the number of figure in the brackets so it would be more clear on which graph this text refer to, and also revise the sentence in order to avoid repetition (K-CSB isotherm).

 Lines 176-178 – equalize C and carbon

 Figure 1 – it is almost invisible and must be enlarged to see all data on it. Otherwise, it is not needed at all, because in this way it doesn’t have purpose.

It can be in two lines instead of one line.

 Table 1 – in Vtot, tot should be in the subscript

 Line 227 – Should MnOX be MnOx?

 Lines 236-250 – The effect of adsorption time should be also included in the text.

Please add explanation and values for adsorption rate in the text, for example for each adsorbent what was the range of min and max R (%), for certain TC concentration, or at least the maximum achieved R for each examined adsorbent. Without these results in the text and with so poor quality of figures, where in Figure 5 all data are completely invisible, reader don’t have complete picture about the obtained results.

Please be so kind and give large figures with all data visible.

 Figure 5 – In Fig. a), c) and e) is presented the dependence of capacity q over time at different TC concentrations, and not the dependence of q over Tc concentrations. It is unclear. Please check the text and results in the figure and correct accordingly or revise in order to be fully clear. What is the optimal time for TC adsorption? This is not discussed in the text which refers to this figure.

 Lines 257-259 – “… the adsorption capacity (qe) demonstrates a general downward trend, initially declining rapidly and then tending to decline gradually.”

This statement is correct but authors should give short explanation for this, short discussion is needed.

 Line 259 – What authors mean by “biochar equilibrium”?

 Line 269 – “Generally, the pH values of TC are 3.3, 7.7, and 9.7 [47]”

What authors mean by this general pH of TC? It is unclear.

 Line 277 – pH instead of PH

 Figure 8 – This figure presents the effect of time. Please add in the figure title initial TC concentration.

What is the difference between time in Figure 5 a)c)e) and here in figure 8?

This must be clarified in the manuscript.

 Line 312 – pseudo second-order model (singular, not plural)

 Table 2

Line 313 – biochar should be in the plural since there are 3 samples

K-CSB should not be in bold. Also, few numbers are not in 4 decimal places so this should be equalized (add one zero).

 Line 317 – Not Figure 9, it is Figure 10. This should be corrected.

 Line 319-320 – “Observing higher R2 values for the Langmuir model indicates that the Langmuir equation is more appropriate for describing the adsorption of TC on K-CSB, P-CSB, and M-CSB.” What does this mean? Please give explanation so the purpose of this isotherm study be complete and clear.

 Line 322-323 – Numbers in these lines are not the same as those in Table 3. Please give the numbers in table 3 also on two decimal places, to be equal.

 Line 331-332 – “Experiments on batch adsorption demonstrated that adsorbents prepared from activated crab shell biochar have superior TC adsorption performance.”

Based on what this is concluded? Again the same question, did adsorption experiments with the original unmodified CSB were done? If yes, where are these results and where is the discussion about them?

 Line 336-337 – “Thus, the entire adsorption process of TC by the three adsorbents.”

Please check this sentence, it seems incomplete.

 Second Figure with number 10 and also final figure in the manuscript – It should be Figure 11. Which part of the text in the manuscript describes this figure? It’s not clear. Please check this.

 5. Conclusions

 Line 356-357 – “The Langmuir and pseudo-second-order models best describe the adsorption of TC by the three adsorbents.”

Meaning what? The sentence should be continued.

 Line 361 – “Antibacterial wastewater”? Please check this term. Maybe the more appropriate would be the wastewater polluted by antibiotics?

Author Response

Manuscript ID: foods-2173468

Title: Mesoporous Activated Biochar from Crab Shell with Enhanced Adsorption Performance for Tetracycline

Dear Reviewers,

We would like to thank for Foods for giving us the opportunity to revise our manuscript. And we also thank the reviewers for their careful read and thoughtful comments on previous draft. Their comments and guidance added a lot to the research and increased its scientific content. Therefore, words cannot express their gratitude for the time and effort they put into evaluating this work. We have carefully taken their comments into consideration in preparing our revision, which has resulted in a paper that is clearer, more compelling and broader. The following summarizes how we responded to reviewers’ comments.

Below is our response to their comments.

Thanks for all the help.

Best wishes,

Jiaxing Sun

Revision-author’s response

To Reviewer B:

Abstract

  1. Line 12 – The abbreviations K-CSB, P-CSB and M-CSB should be mentioned earlier in line 11, in the brackets after the full names, and then they can be used in this form through the manuscript.

1A: Thanks for your suggestion, we have changed it.

  1. Line 15 – is this absorption or adsorption?

2A: That should be adsorption, and we have modified.

  1. Introduction
  2. Line 28 – Is this antibiotic only used for livestock and poultry? It seems like that by reading this sentence. Is it also prescribed for people? If yes, is it mean that humans are capable to completely absorb TC?

1A: Thanks for your insightful suggestion. We have modified this sentence.

  1. Moreover, as mentioned in my general comments, the introduction section should be expanded with information about TC, for example why the tetracycline is chosen among numerous of antibiotics used; add information about consumption of tetracycline, maybe amounts on an annual level…

2A: We added information about TC in the introduction, specifically in lines 36-40.

  1. Lines 50-51 – Please check and revise the sentence. Also, please add/give the reference on which you refer to with this sentence.

3A:We have revised this sentence, specifically in lines 55-56.

  1. Line 71 - …using instead of “of KOH, H3PO4 and KMnO4.”

4A: I am sorry for not being able to revise this comment, because this comment may not be fully presented.

  1. Materials and Methods
  2. Line 79 - 100-mesh sieve – is it the same as mentioned in line 96? If yes, this should be equalized.

1A: Yes, the same sieve is used in both processes. We have revised and used the consistent expression.

  1. Line 86 – Is there a specific and justifiable reason why calcination was done at 800°C? It could be shortly explained in the text.

2A: In the previous pre-experiment, we tested different temperatures (500℃, 600℃, 700℃, 800℃, 900℃) to calcinate crab shell biochar. The specific surface areas of crab shell biochar calcined at different temperatures are as follows: 500℃ (41.16 m2/g), 600℃ (78.11 m2/g), 700℃ (127.38 m2/g), 800℃ (181.79 m2/g) and 900℃ (177.91 m2/g). Among them, the crab shell biochar calcined at 800℃ has the highest specific surface area, so the crab shell biochar calcined at 800℃ is selected for the following experiment in the text.

  1. Line 92 – Please give the information about KOH concentration.

3A: In this study, we used solid KOH, and the samples were mixed by weight ratio.

  1. Line 93 – The CSB was calcined twice at 800°C, before and after modification? Can you shortly explain why?

4A: The first calcination of crab shell is for the preparation of crab shell biochar, detailed in lines 88-93. In the second calcination, crab shell biochar is mixed with solid KOH for high temperature reaction, in order to activate biochar through high temperature reaction of KOH and C, specifically in lines 97-102.

  1. Line 112 – Authors should refer here at certain reference/references in regard to the BET method. In general, the main data about the methods applied for characterisation of crab shell biochar should be given here.

5A: Thanks for your suggestion, we have revised this section, specifically in lines 115-119.

  1. Line 117 – Did you used also unmodified original CSB in adsorption studies? I don’t see these results. It is very important for comparison with chemically modified samples in order to determine is the modification justified (look general comments).

6A: This paper focuses on comparing the differences among the three modification methods, without comparing the original CSB. Thanks for your suggestion, we will analyze the original CSB in the following research.

  1. Lines 126, 137 and 153 – capacity of TC – this must be corrected. It is not capacity of TC; it is capacity of modified CSBs for TC.

7A: Thank you for your correction, we have revised.

  1. Line 143 – in qt the t should be in subscript.

8A: Thanks for your correction, we have changed.

  1. Results and discussion
  2. Line 167 – please give the number of figure in the brackets so it would be more clear on which graph this text refer to, and also revise the sentence in order to avoid repetition (K-CSB isotherm).

1A: Thanks for your suggestion, we have modified the description of this sentence and added the image serial number, specifically in lines 172-178.

  1. Lines 176-178 – equalize C and carbon.

2A: Thank you for correction, we have revised.

  1. Figure 1 – it is almost invisible and must be enlarged to see all data on it. Otherwise, it is not needed at all, because in this way it doesn’t have purpose. It can be in two lines instead of one line.

3A: Thank you for your comments. We have rearranged the images.

  1. Table 1 – in Vtot, tot should be in the subscript.

4A: Thanks for your reminding. We have modified it.

  1. Line 227 – Should MnOX be MnOx?

5A: Thanks for your correction, we have modified it.

  1. Lines 236-250 – The effect of adsorption time should be also included in the text.

6A: As for the influence of adsorption time on the adsorption experiment, we separately put it in 3.2.4, detailed in lines 313-324.

  1. Please add explanation and values for adsorption rate in the text, for example for each adsorbent what was the range of min and max R (%), for certain TC concentration, or at least the maximum achieved R for each examined adsorbent. Without these results in the text and with so poor quality of figures, where in Figure 5 all data are completely invisible, reader don’t have complete picture about the obtained results. Please be so kind and give large figures with all data visible.

7A: Thanks for your comments, we have added the corresponding explanation in lines 256-261.

  1. Figure 5 – In Fig. a), c) and e) is presented the dependence of capacity q over time at different TC concentrations, and not the dependence of q over Tc concentrations. It is unclear. Please check the text and results in the figure and correct accordingly or revise in order to be fully clear. What is the optimal time for TC adsorption? This is not discussed in the text which refers to this figure.

8A: Thanks for your suggestion, we have revised, detailed in lines 266-267.

  1. Lines 257-259 – “… the adsorption capacity (qe) demonstrates a general downward trend, initially declining rapidly and then tending to decline gradually.” This statement is correct but authors should give short explanation for this, short discussion is needed.

9A: Thanks for your comments, we have added the analysis in the paragraph, specifically in lines 273-285.

  1. Line 259 – What authors mean by “biochar equilibrium”?

10A: There is something wrong with our expression. Now we have modified it, specifically in lines 286-288.

  1. Line 269 – “Generally, the pH values of TC are 3.3, 7.7, and 9.7 [47]” What authors mean by this general pH of TC? It is unclear.

11A: Thank you for your comments. Tetracycline antibiotics all contain acidic phenolic hydroxyl, enol hydroxyl and basic dimethylamine. These drugs are amphoteric compounds, with three pKa values of 2.8 ~ 3.4, 7.2 ~ 7.8 and 9.1 ~ 9.7, respectively. The basic group of tetracycline is 4α-dimethylamine, and C-10 phenol hydroxyl group and its conjugate C-12 enol hydroxyl group are weakly acidic groups. We have modified the content of this part, specifically in lines 297-299.

  1. Line 277 – pH instead of PH

12A: Thank you for your correction, we have revised.

  1. Figure 8 – This figure presents the effect of time. Please add in the figure title initial TC concentration.

13A: Thanks for your suggestion, we have added the initial TC concentration in the figure title, specifically in line 326.

  1. What is the difference between time in Figure 5 a)c)e) and here in figure 8? This must be clarified in the manuscript.

14A: Figure 5 a)c)e) is on the adsorption of TC with different initial concentrations by modified biochar, and the adsorption time is 2880 min. Figure 8 is the adsorption effect of the adsorption time on 400 mg/L of TC, and the adsorption time is 3600 min. We have added in lines 315-317.

  1. Line 312 – pseudo second-order model (singular, not plural)

15A: Thank you for your correction. We have modified it.

  1. Line 313 – biochar should be in the plural since there are 3 samples

K-CSB should not be in bold. Also, few numbers are not in 4 decimal places so this should be equalized (add one zero).

16A: Thanks for your suggestion, we have modified this part.

  1. Line 317 – Not Figure 9, it is Figure 10. This should be corrected.

17A: Thanks for your correction, we have revised.

  1. Line 319-320 – “Observing higher R2 values for the Langmuir model indicates that the Langmuir equation is more appropriate for describing the adsorption of TC on K-CSB, P-CSB, and M-CSB.” What does this mean? Please give explanation so the purpose of this isotherm study be complete and clear.

18A: Thank you for your suggestion. We have added the content of this part, specifically in lines 351-357.

  1. Line 322-323 – Numbers in these lines are not the same as those in Table 3. Please give the numbers in table 3 also on two decimal places, to be equal.

19A: Thanks for your suggestion, we have modified.

  1. Line 331-332 – “Experiments on batch adsorption demonstrated that adsorbents prepared from activated crab shell biochar have superior TC adsorption performance.”

Based on what this is concluded? Again the same question, did adsorption experiments with the original unmodified CSB were done? If yes, where are these results and where is the discussion about them?

20A: When 0.05g modified crab shell biochar were added into 50ml 400mg/L tetracycline solution, the final adsorption rate of K-CSB to tetracycline solution could reach 95.05%, P-CSB could reach 82.89%, and M-CSB could reach 70.33%. Based on these results, we indicated that the adsorption effect of modified crab shell biochar on TC solution was better. The purpose of this paper is to discuss the differences between the three modification methods, without discussing the physical and chemical properties of the original CSB. Thank you for your suggestion, and we will continue to study the crab shell biochar in the later study.

  1. Line 336-337 – “Thus, the entire adsorption process of TC by the three adsorbents.” Please check this sentence, it seems incomplete.

21A: Thanks for your suggestion, we have redescribed this part, detailed in line 373.

  1. Second Figure with number 10 and also final figure in the manuscript – It should be Figure 11. Which part of the text in the manuscript describes this figure? It’s not clear. Please check this.

22A: Thanks for your reminding, we have rearranged the figures.

  1. Conclusions
  2. Line 356-357 – “The Langmuir and pseudo-second-order models best describe the adsorption of TC by the three adsorbents.” Meaning what? The sentence should be continued.

1A: Thanks for your suggestion, we have revised the sentence, specifically in line 393-396.

  1. Line 361 – “Antibacterial wastewater”? Please check this term. Maybe the more appropriate would be the wastewater polluted by antibiotics?

2A: Thanks for your suggestion, we have modified this part, specifically in line 400.

Round 2

Reviewer 2 Report

The manuscript is significantly improved which increase its clarity. Authors responded to all questions and comments and make significant modifications through the manuscript.

 Still, there are some lacks that I recommend to be taken into account:

Regarding my comments from the previous revision, and authors answer:

 Line 76 – The KOH, H3PO4 and KMnO4 were used for modification, they are not modified. So instead „of KOH, H3PO4 and KMnO4“ please write „using KOH, H3PO4 and KMnO4“. I believe this is more correct.

 2. Line 86 – Is there a specific and justifiable reason why calcination was done at 800°C? It could be shortly explained in the text.

2A: In the previous pre-experiment, we tested different temperatures (500℃, 600℃, 700℃, 800℃, 900℃) to calcinate crab shell biochar. The specific surface areas of crab shell biochar calcined at different temperatures are as follows: 500℃ (41.16 m 2 /g), 600℃ (78.11 m 2 /g), 700℃ (127.38 m 2 /g), 800℃ (181.79 m 2 /g) and 900℃ (177.91 m 2 /g). Among them, the crab shell biochar calcined at 800℃ has the highest specific surface area, so the crab shell biochar calcined at 800℃ is selected for the following experiment in the text.

 After authors answer is all clear. But, short explanation about this should be added in the manuscript. I don’t see reason why not, when it explains the chosen temperature. If maybe these results have been already published, then give one sentence about them and refer to the certain reference.

I don’t see why numbers in graphs (at y and x axes) are not the same font size as the axes titles? In this way all data in figures will be visible. Figures are improved since the last version, but this suggestion will more increase their visibility.

Also, it is a pity that this paper did not also investigate the original CSB and then focus on comparing the capacity of this material with the modified ones. Definitely it should be done in future research.

Author Response

Manuscript ID: foods-2173468

Title: Mesoporous Activated Biochar from Crab Shell with Enhanced Adsorption Performance for Tetracycline

Dear Reviewer,

We appreciate you very much for your positive and constructive comments and suggestions on our manuscript. We have studied your comments carefully and have made revision which marked in yellow in the paper. We have tried our best to revise our manuscript according to the comments. Attached please find the revised version, which we would like to submit for your kind consideration. We would like to express our great appreciation to you and reviewers for comments on our paper. Looking forward to hearing from you.

Thank you and best regards.

Best wishes,

Jiaxing Sun

Revision-author’s response

  1. Line 76 – The KOH, H3PO4 and KMnO4 were used for modification, they are not modified. So instead “of KOH, H3PO4 and KMnO4” please write “using KOH, H3PO4 and KMnO4”. I believe this is more correct.

1A: Thank you very much for your suggestion. We have revised it, specifically in line 76.

  1. Line 86 – Is there a specific and justifiable reason why calcination was done at 800°C? It could be shortly explained in the text.

A: In the previous pre-experiment, we tested different temperatures (500℃, 600℃, 700℃, 800℃, 900℃) to calcinate crab shell biochar. The specific surface areas of crab shell biochar calcined at different temperatures are as follows: 500℃ (41.16 m2/g), 600℃ (78.11 m2/g), 700℃ (127.38 m2/g), 800℃ (181.79 m2/g) and 900℃ (177.91 m2/g). Among them, the crab shell biochar calcined at 800℃ has the highest specific surface area, so the crab shell biochar calcined at 800℃ is selected for the following experiment in the text.

After authors answer is all clear. But, short explanation about this should be added in the manuscript. I don’t see reason why not, when it explains the chosen temperature. If maybe these results have been already published, then give one sentence about them and refer to the certain reference.

2A: Thank you very much for your comments. We have added the relevant content in section 2.2. Preparation of crab shell biochar and 3.1.1. Porous structure.

  1. I don’t see why numbers in graphs (at y and x axes) are not the same font size as the axes titles? In this way all data in figures will be visible. Figures are improved since the last version, but this suggestion will more increase their visibility.

3A: Thanks for your guidance, we have revised the graphs.
